# Overview of the Role of Pharmacological Management of Obstructive Sleep Apnea

**DOI:** 10.3390/medicina58020225

**Published:** 2022-02-02

**Authors:** Enrique Arredondo, Monica DeLeon, Ishimwe Masozera, Ladan Panahi, George Udeani, Nhan Tran, Chi K. Nguyen, Chairat Atphaisit, Brooke de la Sota, Gabriel Gonzalez Jr., Eileen Liou, Zack Mayo, Jennifer Nwosu, Tori L. Shiver

**Affiliations:** 1Department of Pharmacy Practice, Texas A&M Rangel College of Pharmacy, 1010 W Ave B, Kingsville, TX 78363, USA; enrique_0235@tamu.edu (E.A.); mdeleon@tamu.edu (M.D.); imasozera@tamu.edu (I.M.); udeani@tamu.edu (G.U.); nhtran@tamu.edu (N.T.); chiknguyen@tamu.edu (C.K.N.); atphaisit@tamu.edu (C.A.); brookedew@tamu.edu (B.d.l.S.); gabe.gonzalez23@tamu.edu (G.G.J.); liou.eileen@tamu.edu (E.L.); mayo.z@tamu.edu (Z.M.); jnwosu67@tamu.edu (J.N.); torilynshiver@tamu.edu (T.L.S.); 2Department of Pharmacy Practice, Texas A&M Rangel College of Pharmacy, 59 Reynolds Medical Building, College Station, TX 77843, USA

**Keywords:** obstructive sleep apnea, OSA management, OSA drug therapy, endotypes, phenotypes

## Abstract

Obstructive sleep apnea (OSA) remains a prominent disease state characterized by the recurrent collapse of the upper airway while sleeping. To date, current treatment may include continuous positive airway pressure (CPAP), lifestyle changes, behavioral modification, mandibular advancement devices, and surgical treatment. However, due to the desire for a more convenient mode of management, pharmacological treatment has been thoroughly investigated as a means for a potential alternative in OSA treatment. OSA can be distinguished into various endotypic or phenotypic classes, allowing pharmacological treatment to better target the root cause or symptoms of OSA. Some medications available for use include antidepressants, CNS stimulants, nasal decongestants, carbonic anhydrase inhibitors, and potassium channel blockers. This review will cover the findings of currently available and future study medications that could potentially play a role in OSA therapy.

## 1. Introduction

Obstructive sleep apnea (OSA) is a prominent chronic disease of upper airway obstruction that predominantly affects middle-aged and elderly populations and frequently goes undiagnosed and under-treated [1]. Clinical presentation of OSA includes snoring, sleep fragmentation, episodes of ceased breathing with mid-sleep awakenings, hypoxia, hypercapnia, and interruptive awakenings during sleep [2]. Although the clinical presentation of OSA seems harmless, if left untreated, it can increase the risk of cardiovascular diseases (CVD) such as stroke, hypertension, atrial fibrillation, pulmonary hypertension, and myocardial infarction. This risk is attributed to multifactorial factors such as systemic inflammation, metabolic dysfunction, endothelial dysfunction, and hyperactive sympathetic dysfunction [3,4,5,6,7]. A recent analysis estimated a global rate of 936 million adults living with mild to severe OSA and 425 million adults living with moderate to severe OSA [8]. Treatment methods include continuous positive airway pressure (CPAP), behavioral modification, mandibular advancement devices, and surgical treatment. Although CPAP is the gold standard due to its ability to increase oxygenation in the blood and decrease blood pressure and sleep fragmentation caused by OSA [9,10,11], the short-term patient compliance rate may vary from 42% to 75%, with higher compliance potentially attributed to factors such as increases in patient education and follow-up visits [12,13]. Pharmacological agents such as corticosteroids and nasal saline rinses have demonstrated moderate effectiveness in improving CPAP compliance when in combination with other standard interventions such as mask refitting and heated humidification [14]. Although the aforementioned methods have been shown to improve CPAP compliance, prior studies suggest that up to 24% of patients may not initiate or even fulfill their CPAP prescription due to factors such as the inability to take time off, insurance issues, switch of care to other physicians, and intolerance of CPAP [15]. Due to the desire for a more convenient method of management, pharmacological treatment has been thoroughly investigated as a means for a potential alternative in OSA treatment. More research and resources need to still be dedicated towards pharmacological treatment to prove its efficacy, improve treatment adherence rates, and increase accessibility.

## 2. Endotypes and Phenotypes of Obstructive Sleep Apnea

OSA is a complex disease caused by varying mechanisms that are still not completely understood. Because of this, some researchers and clinicians have begun categorizing OSA as a heterogeneous disease with classifications based on endotypes (underlying mechanisms) or phenotypes (clinical expression) [16,17]. Although OSA classification as a heterogeneous disease has not been fully recognized, other respiratory diseases such as asthma, chronic obstructive pulmonary disease (COPD), and acute respiratory distress syndrome (ARDS) have been recognized as heterogeneous diseases [18,19,20]. The endotypes that can be identified in OSA include impaired anatomical compromise, impaired pharyngeal dilator muscle function, high loop gain, and low arousal threshold, which is the predisposal of waking up due to respiratory disturbances [16,17]. In contrast, OSA phenotypes can be classified into five distinct groups: disturbed sleep, minimal symptoms, upper airway symptoms with sleepiness, upper airway symptoms dominant, and sleepiness dominant [21,22]. Because endotypes are associated with different underlying mechanisms of OSA, adopting this idea of OSA being a heterogeneous chronic disease could be beneficial since it would allow a more patient-specific treatment approach. This review will focus on the different pharmacological treatment options for OSA based on specific endotype classification.

## 3. Upper Airway Anatomic Occlusion or Impaired Anatomy

### 3.1. Weight Loss Medications

Obesity remains the leading risk factor that predisposes or exacerbates OSA in the adult population. It is estimated that obesity is prevalent in approximately 45% of individuals with OSA [23,24,25]. Obesity increases fat deposition around the neck and upper airway, leading to an increased risk of pharyngeal collapse [26,27]. Wang et al. conducted a study to evaluate the effects of weight loss on upper airway anatomy in 67 individuals with OSA (apnea-hypopnea index ^3^10 events/h). Weight loss of the subjects was conducted through lifestyle modifications, bariatric surgery, bypass, or banding. Magnetic resonance imaging (MRI) was utilized to measure the upper airway’s anatomical and fat percentage changes. The authors determined that weight loss led to reduced volumes of upper airway soft tissue and was strongly associated with reductions in apnea-hypopnea index (AHI) [28]. This study demonstrated a strong correlation between obesity and sleep apnea [28]. AHI refers to the number of apneas or hypopneas recorded during the study per hour of sleep. It is used to classify the severity of OSA. Minimal or no OSA is defined as AHI <5 per hour, whereas ≥30 events per hour is considered severe OSA. AHI has been the direct measurement of endotype traits and is also used to determine the efficacy of pharmacologic treatment of OSA [29]. According to the American Thoracic Society (ATS), regarding the weight management of OSA patients, weight-loss medications should be considered adjunctive therapy in those diagnosed with OSA and who are overweight or obese and have been unsuccessful in losing weight with lifestyle interventions [30]. ATS further defines those who will benefit from weight loss medications as patients who have a BMI of 27 kg/m^2^ with weight-related comorbidities such as OSA [30]. Current pharmacological agents that have been approved by the Food and Drug Administration (FDA) for weight loss include orlistat, liraglutide, naltrexone/bupropion, and phentermine/topiramate extended-release (ER). However, only phentermine/topiramate ER, liraglutide, and orlistat have studies evaluating weight loss in those with OSA. A randomized clinical trial studied the efficacy of phentermine 15 mg plus topiramate 92 mg extended-release for moderate to severe OSA. The trial involved forty-five subjects with moderate to severe OSA not receiving the gold standard treatment of continuous positive airway pressure. The primary endpoint was evaluated as AHI and showed a substantial benefit in the phentermine 15 mg in combination with topiramate 92 mg extended-release with lifestyle modifications treatment group compared to the placebo group, which only underwent lifestyle modifications. Primary outcomes of interest were changes in AHI between baseline, week 8, week 28, or withdrawal. Secondary outcomes included respiratory disturbance index, apnea index, hypopnea index, desaturation index, mean overnight oxygen saturation, overnight minimum oxygen saturation, and arousal index. The treatment group’s average number of apnea-hypopnea events was reduced from 44 events/h to 14 events/h, greater than the placebo group at 45 to 27 events/h. In addition to the reduction in AHI, the treatment group had a mean decrease of 10.2% in weight and had a positive, significant correlation between percent change in weight and change in AHI (*p* = 0.0003) [31]. In addition to AHI, the arousal index—which is the sleep quality defined as the number of arousals from REM and NREM sleep per hour—was also studied. The phentermine 15 mg/topiramate 92 mg extended-release group had a least-squared mean decrease in the number of arousals of 19.5 and 21.2 for the placebo group. However, the difference between the groups was not statistically significant. Some limitations noted in the randomized control trial include a limited sample size that may not represent the general population, a short study time frame, and no significant differences between the groups in arousal index, diastolic blood pressure, desaturation index, minimum overnight oxygen saturation, and glycemic and lipid values. The inability to detect major changes in the groups may be attributed to the weight loss experienced by lifestyle modifications in the placebo group [31]. Two randomized control trials have evaluated the effects of liraglutide on weight loss and the correlation to reduction of OSA severity. Blackman et al. evaluated if liraglutide 3.0 mg reduces OSA severity compared to placebo by measuring AHI after 32 weeks of treatment in the SCALE- Sleep Apnea trial. Patients in the treatment and placebo groups received counseling on diet and exercise; however, the treatment group was initiated on 0.6 mg/day and titrated weekly by 0.6-mg increments until a 3.0 mg daily dose. The mean reduction in AHI was greater with liraglutide (−12.2 events/h) than with placebo (−6.1 events/h). The treatment effects on AHI did not depend on the subjects’ OSA severity category, baseline BMI, or gender. (*p* > 0.05 for all subcategories). In addition, liraglutide produced a greater mean percentage weight loss (−5.7%) when compared to placebo (−1.6%) and demonstrated a statistically significant association between weight loss and reduction in OSA endpoints. This study had some major strengths, including a larger sample size compared to other studies looking at the effects of weight loss and improvement in OSA. In addition, polysomnographic (PSG) assessments were conducted, and results were interpreted at specialized sleep sites, leading to increased reliability. However, limitations included being limited to 32 weeks and having 23% of the subjects withdraw across both groups [32]. Overall, these studies demonstrated that the medications used in the trials were useful in weight loss and, consequently, AHI was reduced. Another important thing to note is that the reduction in AHI seen in these studies is not necessarily a direct result of the specific drug being used but instead is more likely a result of the weight loss. Sprung et al. are currently conducting a study designed as a two-by-two factorial design. One hundred and thirty-two patients with newly diagnosed OSA, existing obesity, and type two diabetes mellitus were stratified to receive either liraglutide 1.8 mg, liraglutide 1.8 mg once per day with CPAP, CPAP alone, or no treatment. The study is currently in the follow-up phase, with no preliminary results posted [33]. However, some strengths and limitations were noted in the study. This study has included existing diabetes as an inclusion criterion representing a typical OSA patient [34,35]. The current study protocol limits liraglutide to 1.8 mg once a day, although it could be increased to 3.0 mg daily, which can potentially cause a greater weight loss effect and thus improve OSA. This is because liraglutide 3.0 mg was not yet approved at the time of the study. Currently, there are two randomized control trials that are evaluating the use of orlistat in obese patients. Both are similar in study design, where a hypocaloric diet was initiated followed by a maintenance diet. Rössner et al. conducted a run-in period during the first year, where both groups received a placebo and a nutritionally balanced diet designed to cause a 600-kcal deficit. Patients who finished the run-in period were randomized to receive either 60 mg or 120 mg three times a day during the second year [36]. Similarly, Sjöström et al. conducted a study reflecting the previous study; however, orlistat 120 mg three times a day was utilized [36,37]. OSA outcomes were not measured in the studies. However, weight loss from orlistat may be beneficial, but further studies are needed to assess correlation.

### 3.2. Nasal Decongestants

Another mechanism many have hypothesized in obstructive sleep apnea is impairment of nasal breathing or nasal obstruction [38,39,40]. This has led to speculation that nasal decongestions may improve symptoms of OSA. A randomized, placebo-controlled double-blind crossover study was performed in OSA patients that suffer from chronic nasal obstruction. Patients were recruited after completing overnight studies and were given either placebo or oxymetazoline 0.05% 0.4 mL applied in each nostril. Oxymetazoline, a sympathomimetic vasoconstrictor, was shown to decrease mean AHI in both NREM and REM, 31.65 ± 16.98, compared to placebo at 22.64 ± 16.05. Not only did this show a reduction in OSA severity but oxygen saturation during sleep also improved significantly. Additionally, there was a decreased amount of time spent below 90% oxygen saturation in the treatment group 1% (0.17%, 3.76%) vs. 0.26% (0.00%, 2.15%) in the control group. Several limitations existed in this study, including small sample size and not assessing the functional assessment of the upper airway—more specifically, pharyngeal airway collapsibility [39]. Wijesuriya et al. conducted a prospective, double-blind randomized control trial on patients with OSA who suffered a spinal cord injury. OSA is a common secondary complication in spinal cord injuries or tetraplegia [41,42]. Sleep studies were performed on the study participants in which a nasal spray of 0.5 mL of 5% phenylephrine or placebo was given in random order. Outcomes measured were sleep apnea severity, perceived nasal congestion, sleep quality, and oxygenation during sleep. However, results showed that although nasal resistance was reduced overall, it did not significantly change any sleep severity parameters [41]. More studies are needed in primary OSA patients to determine the efficacy of phenylephrine nasal spray. An additional study assessed the effects of pseudoephedrine and domperidone combination. Patients received one to two capsules containing 60 mg of pseudoephedrine and 10 mg domperidone at bedtime. If the BMI was <28, one capsule was taken; if the BMI was 28–30, one or two capsules were taken, depending on if snoring was present, and; if the BMI was >30, 2 capsules were taken. The results showed a mean decrease of 9.4 (95% CI, 6.8–12.1, *p* < 0.0001) and overall improvement in mean oxygen saturation, percent time with oxygen saturation < 90%, and 4% oxygen saturation desaturation index. However, this study only measured its results using the Epworth Sleepiness Scale score and oximetry measurements. This lack of collecting and evaluating the impact on AHI and improvement in OSA severity posed a major limitation to this study [43].

## 4. Improving Pharyngeal Dilator Function

Serotonin displays predominantly excitatory central effects on 5-HT receptor activity of the upper airway motor neurons and respiratory neurons, specifically 5-HT2a/c and 5-HT1a, respectively. This activity is diminished centrally during REM sleep and is believed to cause upper airway collapse in patients with OSA. Peripherally, serotonin exhibits inhibitory effects involving the 5-HT2a/c and 5-HT3 receptor subtypes [44]. Due in part to their involvement in numerous sites in the central (CNS) and peripheral nervous system (PNS) that control respiration, serotonin reuptake inhibitors (SRIs) have been probed as possible alternative therapies.

### 4.1. Serotonergic Medications

A small double-blind, random-sequence, preliminary study tested the effects of buspirone, a 5-HT1 agonist, in five subjects with OSA conducted by Mendelson et al. Overall, the study resulted in a 36% reduction in AHI against placebo in the group. Buspirone also showed an improvement in sleep quality and an increase in total sleep time. Although the trial was small, with a sample size of five patients, the findings showed the potential use of buspirone in patients with OSA [45]. Berry et al. conducted a double-blind, placebo-controlled crossover study that looked at the effects of selective serotonin reuptake inhibitor (SSRI) and paroxetine 40 mg in eight patients over the course of six weeks. The trial resulted in increased genioglossus activity and muscle responsiveness by 27% in comparison to placebo in non-REM (NREM) sleep in severe OSA patients. Unfortunately, this outcome only occurred on night one, and the AHI did not significantly change. The study presents a limitation to the timing of peak levels after a single dose of paroxetine; the maximum serum level did not occur until late in the night for some of the subjects. Therefore, more extended periods of paroxetine administration are needed to determine if more significant augmentation of genioglossus activity or efficacy in OSA treatment occurs. The study presented some possible confounding effects that could interfere with the study. However, those confounding were well addressed in the study [46]. Hanzel et al. studied twelve patients in a prospective crossover unblinded trial to investigate the effectiveness of fluoxetine in OSA. The group decreased AHI with fluoxetine from a baseline of 57 ± 9 to 34 ± 6. The study also addressed both drugs significantly decreased the proportion of REM sleep time from a baseline pre-drug rate of 17 ± 2% of total sleep time to 3 ± 1% with protriptyline and 7 ± 3% with fluoxetine (F = 8.38; *p* < 0.002). In addition, both drugs decreased the number of apneas or hypopneas in NREM sleep from 58 ± 9 to 32 ± 7 with fluoxetine and to 32 ± 8 with protriptyline (F = 3.31; *p* = 0.05). Thus, fluoxetine shows a potential benefit in OSA; however, the study demonstrates that the dosage used in neither fluoxetine nor protriptyline was uniformly effective in resolving sleep-disordered breathing. Furthermore, both agents’ consequences of intermittent hypoxemia and interrupted sleep significantly reduced the number of apneas or hypopneas, with fluoxetine being more effective and superior to protriptyline agents. The limitation of this study includes the tolerability of more than 10 mg/day of protriptyline. Blood levels of the drugs were not obtained due to the uncertainty that the effects of tricyclic antidepressants are related to their concentration in the blood. The conclusion was only assumed that both dugs were acting on the CNS and could potentially alter sleep-disordered breathing. Age ranged from 25–65 years, with a mean of 51 ± 3 years (±SE). Bodyweight ranged from 66 to 159 kg, with a mean of 99 ± 7 kg [47]. A randomized, double-blind, placebo-controlled subsequent trial tested the combination of fluoxetine and ondansetron, 5-HT_3_ receptor antagonists, in 44 patients with OSA. The result was a 40% reduction in AHI in the high-dose combination group (fluoxetine 10 mg daily and ondansetron 24 mg daily; *p* < 0.03) compared to baseline, which was predominantly attributed to fluoxetine use as the ondansetron monotherapy group showed a trend toward increased AHI post-treatment that was not statistically significant in REM and supine sleep. However, the mean arousal index was higher in the placebo group than in the active treatment group. The study suggested that a high-dose combination treatment may yield an entirely therapeutic response in only a subset of patients with mild-moderate OSA. Thus, fluoxetine may have some benefits as a pharmacological option for OSA [48]. Furthermore, several NREM and REM sleep studies have shown that serotonin SRIs have a strong association with reducing AHI in OSA severity. Although the literature found several limitations and confounding, there is potential evidence for the role of serotonin in seizure control [49]. Thus, there have been noted positive findings in the literature. Larger prospective, phase III trials are still needed to confirm that the improvements observed in OSA are due to selective serotonin receptor inhibitor (SSRI) use.

### 4.2. Noradrenergic Medications

The decrease in noradrenergic activity results in increased genioglossus activity. The tricyclic antidepressant (TCA) protriptyline was conducted in three randomized, double-blind trials. Brownell et al. studied five obese men with severe OSA who were administered protriptyline 20 mg [50]. Although there was no change in AHI and weight loss as a confounding factor, the result from the study implied that patients had improvements in daytime sleepiness and oxygen saturation, as evidenced by the REM reduction during the treatment from 0.231 ± 0.031 to 0.107 ± 0.013 (mean ± S.E.M.) (*p* < 0.05) [50]. Protriptyline 20 mg was also administered for 14 days in 10 patients with moderate-to-severe OSA. A similar subsequent trial by Hanzel et al. demonstrated a 42% reduction in AHI. These results indicate the possible use of protriptyline in particular groups of OSA patients that prefer non-aggressive management [44]. Small patient samples, flawed methodology, and insufficient evidence for the treatment of sleep apnea or sleep-disordered breathing are the current limitations of protriptyline use in OSA patients. The typical dose of protriptyline administered in most studies with pharmacologic intervention is 10 mg or 20 mg PO once daily. Anticholinergic side effects such as dry mouth, urinary retention, and nervousness are common side effects of protriptyline and are a significant reason for drug discontinuation [47]. A randomized, double-blind, placebo-controlled crossover trial studied 20 patients with OSA. Patients were given 80 mg of atomoxetine, a selective norepinephrine reuptake inhibitor (SNRI), and 5 mg of oxybutynin, a synthetic muscarinic antagonist, 30 min before bedtime over two nights. These events resulted in a 50 (± 75%) AHI reduction of 15 ± 22.5 events/h from 30 events/h. The studies were conducted in both REM and NREM stages, with limitations including evaluation of baseline subjective or objective sleepiness, sleep consolidation, and the one-dose regimen. It remains unknown if lower doses of the combination of atomoxetine–oxybutynin could have similar efficacy, fewer side effects, or a higher safety profile [51]. This study was compared with a similar trial that, using desipramine, discovered that it significantly improves upper airway collapsibility. In addition, the study also shows the most significant reduction in AHI on desipramine with minimal upper airway response at baseline. However, despite these improvements, the arousal threshold was reduced on desipramine, which could be the reason for OSA worsening in some patients [52].

### 4.3. Potassium Channel Blockers

Potassium channel blockers have recently been identified as a potential pharmacological treatment for OSA. Monoamine withdrawal can reduce upper airway muscle activity. Potassium channel blockers were found to play a role in the motor neuron excitability of these monoamines in REM and NREM sleep. This hypothesis was tested in a single-blinded, randomized control trial using 10 mg of 4-Aminopyridine (4-AP), a potassium channel blocker and an orphan drug, in 10 healthy patients. 4-AP or placebo was given 3 h before bed in a controlled environment. This study showed an increase only in the tonic activity of the electromyography of the genioglossus muscle during REM sleep (*p* = 0.04). The restricted effect seen in the study was due to subtherapeutic dosing; however, upper dose titration was not possible due to this drug having a narrow therapeutic index [53].

## 5. High Loop Gain

Loop gain is quantified as the instability of ventilatory chemoreflex control [54]. Loop gain functions as the system that controls ventilation via a negative feedback loop [54]. This mechanism manages blood gas tension levels between narrow limits. The feedback loop consists of various components (i.e., the circulatory delay, the plant, and the controller) and works consecutively to prevent any shift in blood gas tension due to ventilation [54]. A high loop gain signifies the unbalanced response to minor changes in PCO2, leading to hyperventilation mixed with hypoventilation [54]. Patients with a collapsible upper airway and high loop gain experience negative inspiratory pressures that overwhelm the airway and force it closed [54]. Because of this, there has been a growing interest in discovering possible OSA treatments that work through mechanisms that alter ventilatory chemoreflex control [54].

Carbonic anhydrase inhibitors, such as acetazolamide, work by preventing the breakdown of carbonic acid. This results in the accumulation of carbonic acid in the body and acidification of blood pH, resulting in the accumulation of carbonic acid which prompts the kidneys to begin secreting sodium, bicarbonate, chloride, and water in the urine. The clinical result is decreased blood pressure and metabolic acidosis [55,56]. Acetazolamide demonstrates a decrease in resting PCO2 by creating a brief metabolic acidotic state and possible hyperventilation. Lowering PCO2 decreases the chance for shifts in ventilation [44].

### Carbonic Anhydrase Inhibitors

1000 mg (250 mg QID) of acetazolamide was studied in 10 patients with moderate-to-severe OSA in a randomized, placebo-controlled, crossover trial by Whyte et al. over two weeks [44]. The inclusion criteria for this study were patients with more than 15 apneas and hypopneas per hour of sleep, along with symptoms of persistent nocturnal awakening, loud snoring, inadequate nocturnal sleep, and daytime sleepiness. The results of this study showed that in nine patients, acetazolamide stimulated remarkable metabolic acidosis with a mean pH of 7.42 ± 0.04 seen with placebo and 7.36 ± 0.04 seen with acetazolamide (*p* < 0.01). Additionally, there was an overall reduction in end-tidal PCO2 (baseline of 40.2 ± 0.8 to 32.5 ± 1.0) and a 52% reduction in AHI compared to placebo. While these may seem to be positive outcomes, no changes were noted in the number of sleep arousals per hour (26 ± 26 placebo, 16 ± 10 h of sleep acetazolamide, *p* > 0.2), nor in the previous symptoms mentioned. Moreover, a REM analysis was not conducted [57]. Another study from Edwards et al. involving 12 non-naive OSA subjects with concurrent CPAP therapy and an AHI of more than 10/h of sleep were given 500 mg twice daily. Patients had no significant difference in their BMI or neck circumference. Acetazolamide reduced the loop gain among the group by 41% (*p* < 0.05), proving its effectiveness in loop gain. AHI was recorded during REM and NREM sleep; however, OSA traits were collected during NREM sleep, only causing correlations specific to NREM sleep [56]. However, despite these promising results, there was no clinically significant improvement in symptoms such as daytime sleepiness. Side effects common with acetazolamide use are paresthesia and nocturia [55,56,57].

## 6. Low Respiratory Arousal Threshold

In patients with OSA, sleep arousal is the sudden awakening from sleep due to airway obstruction. Approximately 30–50% of OSA patients have low respiratory arousal thresholds, making them more subject to frequent and premature nighttime awakenings, leading to sleep fragmentation, breathing instability, and impaired dilator muscle [58,59]. Because of this, further efforts have been made to investigate drugs with sedative and hypnotic effects with the intent of increasing arousal threshold while maintaining appropriate upper airway muscle activity, to reduce AHI and OSA severity.

### 6.1. Benzodiazepines

Benzodiazepines produce sedative, hypnotic, myorelaxant, and anticonvulsive effects by working on GABA_A_ receptors and increasing inhibitory neurotransmitters in the CNS [60]. Because of this, benzodiazepines were once contraindicated in patients with OSA for fear of worsening symptoms. However, a more recent discovery of OSA phenotypes has shifted this view of benzodiazepines and their ability to slow arousal responses to one beneficial in OSA patients with low arousal thresholds. Berry and colleagues conducted two randomized, double-blind crossover studies and found that triazolam (0.25 mg) significantly increased arousal thresholds by approximately 33% in healthy individuals and 24% in severe OSA patients [61,62]. Notably, both Berry’s studies were conducted in males. While the first study investigated the effect of triazolam 0.25 mg on arousal response to airway occlusion during non-REM sleep in young group males (mean age ± SD, 28.1 ± 7.1 yr), the latter one examined the middle-aged male with a mean (±SD) age of 46.6 ± 14.1 yr and severe OSA. Another study by Hoijer and colleagues looked at the effect of nitrazepam (5 or 10 mg) compared to placebo in patients with mild to moderate obstructive sleep apnea and found that although it did increase arousal threshold, it had no effect on AHI and did not improve OSA severity. However, nitrazepam can modestly increase total sleep time and decrease REM sleep. Similar to Berry’s studies, U Hoijer was conducted only on males [63]. One study looked at the effects of different hypnotics on individuals with and without OSA and found that temazepam 10 mg showed no significant effect on respiratory arousal threshold or upper airway muscle activity compared to placebo. However, the effects on upper airway muscle activity within the group did show that some individuals were more affected than others, demonstrating that the use of benzodiazepines is not favorable for all OSA patients. Overall, the results of these studies show that although benzodiazepines mainly work as predicted, by increasing arousal threshold, this effect does not consistently correlate to improved AHI and OSA severity. Therefore, future studies are needed further to assess the clinical implications of benzodiazepine use in OSA therapy.

### 6.2. Z-Drugs

Z-drugs are non-benzodiazepine agents that work on GABA_A_ receptors to produce inhibitory effects. However, unlike benzodiazepines, z-drugs do not produce myorelaxant effects and are associated with fewer adverse side effects, which their increased selectivity could explain GABA_A_$\alpha_{1}$ subunits interaction [60]. In a physiological study, eszopiclone (3 mg) was found to significantly increase the arousal threshold by approximately 30% (*p* < 0.01) from stage one to stage two sleep and reduce AHI by about 45% (*p* = 0.52) in OSA patients with a low arousal threshold. Although the result sounded very promising, the sample size in this study was small. Therefore, the study suggested a larger clinical trial to confirm eszopiclone’s effect in manipulating the arousal threshold. Additionally, this study showed that eszopiclone effects were more variable in those with higher arousal thresholds and less beneficial in those patients with poor muscle responsiveness [58]. Another study looked at the effects of zolpidem 10 mg and zopiclone 7.5 mg in patients with and without OSA. Results showed that zolpidem and zopiclone increased the arousal threshold by 27% (*p* = 0.02) and 37% (*p* < 0.001), respectively, without inhibiting upper airway muscle response [58]. However, a different study looked at the effects of zolpidem (10 mg) in patients with severe OSA and low-moderate arousal threshold and found that although arousal threshold increased by 15% (*p* = 0.010), it was not shown to reduce AHI and improve OSA severity [64].

### 6.3. Other Altzernative Hypnotics

Trazadone, a tricyclic antidepressant, is another drug that is being investigated for OSA patients with a low arousal phenotype because of the sedative effects it can produce by blocking muscarinic a-adrenoceptors [60]. A study looked at the effect of trazodone 100 mg in OSA patients with a low arousal threshold and found that it significantly increased the respiratory arousal threshold by 32% without altering upper airway muscle activity. Despite these findings, no significant reduction in AHI and OSA severity was noted [65]. However, another study did find that trazodone (100 mg) significantly reduced AHI when compared to placebo (38.7 vs. 28.5 events/h, *p* = 0.041) in OSA patients with AHI ≥ 10, yet the study denied trazodone’s effect on the non-REM arousal threshold compared with placebo [66].

In subgroup analysis, responders to trazodone presented more stable breath that meditated shorter N1 sleep than placebo (20.1% placebo vs. 9.0% trazodone, *p* = 0.052), and decreased arousal index non-responders were not observed to have a change in sleep parameters.

These findings show that the prior belief to avoid sedatives in all OSA patients, due to the risk of upper airway muscle activity impairment, is more accurate for OSA patients who have higher arousal thresholds and impaired upper airways muscle responses. OSA patients who express a phenotype of low arousal threshold and who have proper functioning upper airway muscles could benefit from sedatives. Z-drugs and tricyclic antidepressant trazodone have shown to be effective in increasing respiratory arousal threshold and reducing AHI, whereas benzodiazepines have shown little benefit in reducing OSA severity. Overall, these studies show that the use of sedatives in certain OSA patients is not harmful and, in some cases, may even be beneficial. Because of this, future studies to further investigate potential roles for sedative use in OSA therapy would be of great value in the advancement of OSA treatment. The mechanism of action for the drugs discussed above, divided according to their endotypic characterization, are summarized in Table 1.

## 7. Future Studies and Conclusions

Expansion of pharmacological, non-mechanical therapy for OSA is on the rise and can contribute as a non-invasive option for patients. Table 2 highlights the current clinical trials in progress or have recently been completed on pharmacological options in OSA [68].

Due to CPAP side effects, discomfort, lack of knowledge of CPAP’s effects, and subjective sleep-related symptoms may contribute to inconsistent adherence. The limitations of CPAP make pharmacological agents an advantage as medications are more convenient and an alternative option for patients [69,70]. Future research will potentially solidify the role of pharmacological agents in the management of OSA.

## Figures and Tables

**Table 1 medicina-58-00225-t001:** Potential Medication Options and Proposed Mechanism of Action.

Endotype	Drug Examined	Mechanism of Action	Dosages That Were Studied	References
Upper airway anatomic occlusion or impaired anatomy	Liraglutide	GLP-1 agonist increases insulin secretion, decreasing glucagon secretion and slowing gastric emptying, leading to weight loss	1.5–3.0 mg once a day	[32,33]
Upper airway anatomic occlusion or impaired anatomy	Phentermine/topiramate	Phentermine: reduces appetite through the activation of sympathomimetic amines Topiramate: Increased GABA activity, blocks neuronal voltage-dependent sodium channels and antagonizes AMPA glutamate receptors	Phentermine: 15 mg and Topiramate extended-release: 92 mg once a day	[31]
Upper airway anatomic occlusion or impaired anatomy	Orlistat	Inhibitor of gastric and pancreatic lipases, decreasing the absorption of dietary fats and leading to weight loss	60–120 mg three times a day	[36,37]
Upper airway anatomic occlusion or impaired anatomy	Oxymetazoline	Stimulates α-adrenergic receptors in the arterioles of the nasal mucosa leading to vasoconstriction. Activation of α-adrenergic receptors leads to decreased nasal patency	0.05%, 0.4 mL each nostril at bedtime and 3 h after	[39]
Upper airway anatomic occlusion or impaired anatomy	Phenylephrine	Stimulates α-adrenergic receptors in the arterioles of the nasal mucosa, leading to vasoconstriction. Activation of α-adrenergic receptors leads to decreased nasal patency	5%, 0.5 mL each nostril at bedtime	[41]
Upper airway anatomic occlusion or impaired anatomy	Pseudoephedrine/Domperidone	Stimulates α-adrenergic receptors in the arterioles of the nasal mucosa, leading to vasoconstriction. Activation of α-adrenergic receptors leads to decreased nasal patency Domperidone: peripheral dopamine receptor blocking	BMI < 28: 1 capsule of 60 mg pseudoephedrine and 10 mg Domperidone BMI 28–30: 1–2 capsules; depends if snoring is present BMI > 30: 2 capsules	[43]
Improving pharyngeal dilator function	Buspirone	5-HT1 agonist	20 mg/day	[45]
Improving pharyngeal dilator function	Paroxetine	Selective Serotonin Reuptake Inhibitor (SSRI)	40 mg, 4 h before bedtime	[46]
Improving pharyngeal dilator function	Fluoxetine	Selective Serotonin Reuptake Inhibitor (SSRI)	20 mg/day	[47]
Improving pharyngeal dilator function	Protriptyline	TCA increases the synaptic concentration of serotonin and norepinephrine in the CNS	20 mg orally at bedtime for two weeks	[50]
Improving pharyngeal dilator function	Atomoxetine/Oxybutynin	Atomoxetine: Increases norepinephrine concentration by inhibiting reuptake Oxybutynin: Blocks muscarinic receptors on smooth muscle	80 mg, 30 min before bedtime, over two nonconsecutive nights 5 mg, 30 min before bedtime, over two nonconsecutive nights	[51]
Improving pharyngeal dilator function	Desipramine	TCA increases the synaptic concentration of serotonin and norepinephrine in the CNS	200 mg orally at bedtime	[52]
Improving pharyngeal dilator function	4-Aminopyridine	K+ channel blocker	10 mg extended-release daily	[53]
High loop gain	Acetazolamide	Carbonic anhydrase inhibitor, leading to bicarbonate excretion and metabolic acidosis, consequently stimulating baseline ventilation	250 mg four times a day 500 mg twice a day	[44,56]
Low Respiratory Arousal Threshold	Triazolam	Binds to benzodiazepine receptors at the postsynaptic GABA neuron, leading to increased chloride influx hyperpolarizing the cell	0.25 mg, 30–90 min before bedtime	[61,62]
Low Respiratory Arousal Threshold	Nitrazepam	Binds to benzodiazepine receptors at the postsynaptic GABA neuron, leading to increased chloride influx hyperpolarizing the cell	5–10 mg before bedtime	[63]
Low Respiratory Arousal Threshold	Temazepam	Binds to benzodiazepine receptors at the postsynaptic GABA neuron, leading to increased chloride influx hyperpolarizing the cell	10 mg	[67]
Low Respiratory Arousal Threshold	Eszopiclone	Z-drug	3 mg	[58]
Low Respiratory Arousal Threshold	Zolpidem	Z-drug	10 mg	[58,64]
Low Respiratory Arousal Threshold	Zopiclone	Z-drug	7.5 mg	[58]
Low Respiratory Arousal Threshold	Trazodone	Trazodone exhibits 5-HT2A and α1-adrenergic antagonist activity as well as weak serotonin reuptake inhibitor (SSRI)activity	100 mg before sleep	[65,66]

**Table 2 medicina-58-00225-t002:** Current Clinical Trials Involving Medication Management of Obstructive Sleep Apnea Have Recently Been Completed or Ongoing.

Clinical Trial Name	Drug Being Studied	Mechanism of Action	Primary Outcomes Being Studied
A Clinical Pharmacology Study of TS-142 in Patients with Obstructive Sleep Apnea-Hypopnea	TS-142	Unknown	The least-square mean difference of AHI
AD109 Dose Finding in Mild to Moderate OSA	AD109	Targets neurological control and facilitates the activation of the upper airway dilator muscles to maintain an open airway during sleep	Change in hypoxic burden
Study on the Safety of Drug BAY2586116 and How it Works in Patients with Obstructive Sleep Apnea Including the Blood Level of the Drug and Effect of Its Doses and Routes of Administration	Bay2586116	Blocks protein channels expressed on the surface of the upper airways in small mechanoreceptors	Critical closing pressures of the upper airway during sleep with a polysomnography
Optimal Dosage of Acetazolamide for OSA Treatment	Acetazolamide	Carbonic anhydrase inhibitor leading to bicarbonate excretion and metabolic acidosis, consequently stimulating baseline ventilation	AHI
Combination Drug-Therapy for Patients with Untreated Obstructive Sleep Apnea (RESCUE-Combo)	A multidrug combination of Acetazolamide, Eszopiclone, and Venlafaxine	Acetazolamide: Carbonic anhydrase inhibitor leading to bicarbonate excretion and metabolic acidosis, consequently stimulating baseline ventilation Eszopiclone: Z-drug Venlafaxine: serotonergic	AHI during supine non-rapid eye movement sleep
Spironolactone to Improve Apnea and Cardiovascular Markers in Obstructive Sleep Apnea Patients	Spironolactone	Possibly decrease peri pharyngeal fluid accumulation that predisposes individuals to upper airway obstruction	Change in AHI
Study for Efficacy and Dose Escalation of AD313 + Atomoxetine (SEED)	Atomoxetine AD313	Atomoxetine: Increases norepinephrine concentration by inhibiting reuptake AD313: Unknown	AHI from baseline as compared to the highest dose of AD313 at the 28-day polysomnogram
Pharmacological Activation of Hypoglossal Motor Nucleus for Obstructive Sleep Apnea	LTM1201AZ LTM1201AT LTM1201AG LTM1201AD	Hypoglossal motor nucleus activator, leading to increased upper airway dilator muscle activity	AHI
Parallel Arm Trial of AD109 and AD504 in patients with OSA (MARIPOSA)	AD109 + AD504 Atomoxetine hydrochloride	AD109: Unknown AD504: Unknown Atomoxetine: Increases norepinephrine concentration by inhibiting reuptake	AHI for combined AD109 dose arms vs. combined placebo arms
A Novel Pharmacological Therapy for Obstructive Sleep Apnea	Atomoxetine + Oxybutynin	Atomoxetine: Increases norepinephrine concentration by inhibiting reuptake Oxybutynin: Blocks muscarinic receptors on smooth muscle	Change in AHI from baseline
Effect of Oxymetazoline Hydrochloride in Combination with Fluticasone Propionate on the Apnea-Hypopnea Index (AHI) in Subject with Persistent Nasal Congestion and Mild Obstructive Sleep Apnea	Addition of Oxymetazoline Hydrochloride to optimal doses of intranasal fluticasone propionate	Stimulates α-adrenergic receptors in the arterioles of the nasal mucosa, leading to vasoconstriction. Activation of α-adrenergic receptors leads to decreased nasal patency	The proportion of subjects demonstrating a 50% reduction in the apnea-hypopnea index after treatment with oxymetazoline hydrochloride and fluticasone propionate for two weeks
Benefits of Oxytocin in Obstructive Sleep Apnea (OSA) Patients Using Continuous Positive Airway Pressure (CPAP) Machine	Oxytocin	Unknown	Change in CPAP pressures
Combination Pharmacological Interventions for Multiple Mechanisms of Obstructive Sleep Apnea (ComboPlus)	Drug combinations with the following: SAS0421a SAS0421b SAS0421c	Unknown	Change in AHI from baseline
Trial of AD113 and Atomoxetine in OSA Patients with Hypertension	AD113Atomoxetine	AD113: Unknown Atomoxetine: Increases norepinephrine concentration by inhibiting reuptake	Change in hypoxic burden (HB) for AD113 vs. atomoxetine

## Data Availability

Not applicable.

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
