# Peer review of "Overview of the Role of Pharmacological Management of Obstructive Sleep Apnea"

_medicina, 2022, doi:10.3390/medicina58020225_

Round 1

Reviewer 1 Report

Although I believe that this paper was written with a good faith effort, and it is organized logically, the contents have inconsistent dosage references for all the medications cited as possible interventions for OSA. All studies cited should have standard reference to medication dosages, when given, and whether they produce beneficial effects, no change or deleterious effects.

The introductory paragraph, line 5 (or margin reference 32), which cited OSA with narcolepsy as its feature is an error. Sleepiness and sleep attacks may cause patients to think they suffer from narcolepsy, the two disorders should not be confused in a scientific paper. More research would show they are different, though have coinciding features.

Margin #37/38 quoted the estimated global rate (prevalence?) of mild-moderate OSA without citation.

Typically, scientific evidence and the strength of the papers reviewed have an "evidence hierarchy" demonstrating how the authors view the literature they have reviewed. Please incorporate into the article.

There is no mention, if reported, if there is any different effects of medications with these trials on gender, age or at risk populations beside their effects on AHI. There is also no report of exact dosages used in cited trials or timing of use. There is no mention of effect of timing of sleep. Do the medications affect Non-REM sleep? REM sleep? Neither?

Author Response

Reviewer Comment: Although I believe that this paper was written with a good faith effort, and it is organized logically, the contents have inconsistent dosage references for all the medications cited as possible interventions for OSA. All studies cited should have standard reference to medication dosages, when given, and whether they produce beneficial effects, no change or deleterious effects.

Response: Since there is minimal literature on different pharmacological medications for OSA, there are mixed trials using different medication dosing for their subject population. E.g. The liraglutide study conducted by Blackman and colleagues assessing if weight loss is beneficial in those with OSA used a max dosing of 3.0mg/day while Sprung and colleagues used a max dosing of 1.8 mg/day. Medication dosages were further emphasized and are now referenced within Table 1 along with respective references of the study/studies.

Reviewer Comment: The introductory paragraph, line 5 (or margin reference 32), which cited OSA with narcolepsy as its feature is an error. Sleepiness and sleep attacks may cause patients to think they suffer from narcolepsy, the two disorders should not be confused in a scientific paper. More research would show they are different, though have coinciding features.

Response: Thank you for the distinction and it is agreed upon that narcolepsy and OSA are two distinct disorders. “Narcolepsy” was removed as a clinical presentation.

Reviewer Comment: Margin #37/38 quoted the estimated global rate (prevalence?) of mild-moderate OSA without citation.

Response: Citation was added to the corresponding statistic of OSA global prevalence.

Reviewer Comment: Typically, scientific evidence and the strength of the papers reviewed have an "evidence hierarchy" demonstrating how the authors view the literature they have reviewed. Please incorporate into the article.

Response: Thank you for the insight. We have included notable limitations, strengths, and strength of trials within the article.

Reviewer Comment: There is no mention, if reported, if there is any different effects of medications with these trials on gender, age or at risk populations beside their effects on AHI. There is also no report of exact dosages used in cited trials or timing of use. There is no mention of effect of timing of sleep. Do the medications affect Non-REM sleep? REM sleep? Neither?

Response: Referenced articles were analyzed further if sub-group analysis were performed on gender, severity of OSA, risk populations etc. The dosages of medications used in the clinical trials were incorporated in to the paper and in Table one along with reference of studies. Furthermore, effects of AHI in Non-REM, REM or neither were also listed.

Reviewer 2 Report

The presented manuscript raises an important topic about the management of a significant disease such as OSA. As we all well know, the gold standard treatment, CPAP, can be challenging for the patients and has a quite low level of compliance, thus the study of novel treatments is essential.

In general:

Lots of typos and English should be considerably revised.

The review was conducted not following the PRISMA guidelines. The authors have not specified in the paper inclusion and exclusion criteria for the review. Also, databases, registers, websites, organizations, reference lists and other sources searched or consulted to identify studies are not cited and there is no flow diagram.

Abstract:

Line 16. I would cite CPAP as the first treatment for OSA.

Line 19. I would not say that OSA is TYPICALLY classified into endotype and phenotype. This is a proposed classification from some authors.

Keywords: I would have used different keywords adding, for example, OSA management, OSA therapy.

Introduction:

Line 38. As previously said, I would put CPAP first.

Line 47: I would add: …increase accessibility … “and to prove its efficacy”

Endotypes and phenotypes:

Line 50: I would say SOME researchers and not researchers. And please cite.

Line 64. I would delete “current or future”.

Upper airway anatomic occlusion:

Line 74. Delete (Pearson’s rho=….)

Line 75. “Apneas or hypopneas” should be not underlined.

Line 88: Specify the acronym FDA

Line 105. Authors should say that all the cited studies demonstrated that the medications used in the trials were useful in the weight loss and SO the AHI decreased. It should be underlined that the effect on the AHI could be due to the reduction of the weight and not directly to the drug.

Line 105. Studies should have the same outcome to be compared (e.g. AHI).

Line 124. In addition to an improvement in OSA severity?????

Line 127. Please explain RCT acronym.

Line 136. I would delete: (95% CI…)

Low respiratory arousal threshold

Line 309: Table 1 must be cited in the text.

In table 1, column 1 It seems to me the authors are talking about endotype and not phenotype.

Reviewer 3 Report

The review manuscript covers the pharmacological treatment of obstructive sleep apnea syndrome (OSAS). The topic is important, because of the widespread pathophysiology of the disease.

Despite the good summary of the pharmacological treatments, some modification must be made according to the basic, clinical aspects of sleep apnea syndrome.

Narcolepsy is a well defined disease, and  it is not part of the OSAS. Of course, overlap can be seen sometimes (rarely), but this description is confusing. Please, correct it!

Instead of the excessive daytime sleepiness, the sleep fragmetation terminology must be used as cardinal symptom of OSAS. Many patient with OSAS does not suffer from sleepiness, but the consequences of sleep fragmentation can be seen and detected. Please, correct it!

According to the work of B.T. Keenan and coworkers (SleepJ, 2018.1-14, doi: 10.1093/sleep/zsx214)  nowadays five subtipes of OSAS is known. Beside the mentioned three we can speak about additional two forms.  Please, correct it, and insert the reference!

The manuscript containes the description of extremely low compliance for CPAP usage. With correct examninations and follow up the compliance is higher. Relevant information about the real compliance can be find! Please correct it!

In the discussion, it would be good to mention that the pharmacological treatments are capable to increase the compliance of the CPAP usage, but in most cases insuffitient to normalize the apnoe-hypopnoe index alone.

Round 2

Reviewer 2 Report

I ’m satisfied about the corrections that, in my opinion, have significantly improved the manuscript.